# Review of RM-1929 Near-Infrared Photoimmunotherapy Clinical Efficacy for Unresectable and/or Recurrent Head and Neck Squamous Cell Carcinoma

**DOI:** 10.3390/cancers15215117

**Published:** 2023-10-24

**Authors:** Nanami L. Miyazaki, Aki Furusawa, Peter L. Choyke, Hisataka Kobayashi

**Affiliations:** Molecular Imaging Branch, National Cancer Institute, National Institutes of Health, Bethesda, MD 20892, USA; miyazakinl@gwmail.gwu.edu (N.L.M.); aki.furusawa@nih.gov (A.F.); pchoyke@mail.nih.gov (P.L.C.)

**Keywords:** RM-1929, photoimmunotherapy, cetuximab-IR700, cetuximab sarotalocan sodium, recurrent head and neck squamous cell carcinoma (HNSCC)

## Abstract

**Simple Summary:**

RM-1929 near-infrared photoimmunotherapy (NIR-PIT) is an emerging treatment that is currently being investigated under a world-wide Phase III clinical trial and has been conditionally approved for the treatment of unresectable and/or recurrent HNSCC in Japan since 2021. Disease control rates ranged from 66.7 to 100% and overall response rates ranged from 43.3 to 100%. Low-grade postoperative localized pain and edema were the most frequently reported side effects. These preliminary data in real-world use of RM-1929 NIR-PIT show that it is a well-tolerated therapy that has clinically meaningful outcomes for patients with unresectable and/or recurrent HNSCC.

**Abstract:**

Head and neck squamous cell carcinoma (HNSCC) contribute to a significant global cancer burden. Developments in current therapeutic approaches have improved patient outcomes but have limited efficacy in patients with unresectable and/or recurrent HNSCC. RM-1929 near-infrared photoimmunotherapy (NIR-PIT) is an emerging treatment that is currently being investigated in a Phase III clinical trial and has been conditionally approved for the treatment of unresectable and/or recurrent HNSCC in Japan. Here, we collect a series of case reports and clinical trial data to assess the efficacy of RM-1929 NIR-PIT. Disease control rates ranged from 66.7 to 100% across these studies, and overall response rates ranged from 43.3 to 100%, suggesting positive clinical outcomes. Low-grade postoperative localized pain and edema were the most frequently reported side effects, and preliminary reports on quality of life and pain levels suggest that RM-1929 NIR-PIT does not significantly decrease quality of life and is manageable with existing pain management strategies, including opioids. These preliminary data in real-world use of RM-1929 NIR-PIT show that it is a well-tolerated therapy that has clinically meaningful outcomes for patients with unresectable and/or recurrent HNSCC.

## 1. Introduction

Globally in 2020, head and neck squamous cell carcinoma (HNSCC) was the seventh most common cancer, comprising approximately 5% of new cancer cases and deaths [1]. Head and neck cancer tends to affect males, with approximately a 3:1 male-to-female ratio [1], and is projected to be the eighth most common cancer in males in the United States in 2023 [2]. While tobacco-associated cancer is declining, human papillomavirus (HPV)-related HNSCC is rising in incidence and is projected to continue to increase by 1–2% per year [2,3].

Current treatments for HNSCC include surgery, radiation therapy, chemotherapy or a combination of these therapies [4]. Most recently, molecular targeting and immune-activating therapies including cetuximab, cisplatin, and immune-checkpoint inhibitors have also been available for treating HNSCC. Advancements in these treatment modalities have improved outcomes for patients with primary HNSCC. Clinical studies investigating HPV-positive oropharyngeal cancer treatment have demonstrated better clinical outcomes with cisplatin–radiotherapy combination therapy compared with cetuximab–radiotherapy combination therapy. While no significant differences were observed in drug toxicity events, 2-year and 5-year progression-free survival rates were 7–11% higher with cisplatin–radiotherapy treatment, supporting utilizing cisplatin–radiotherapy concomitant therapy as the standard of care for HPV-positive oropharyngeal cancer [5,6]. Although 5-year survival rates in early stage or low-risk HPV-positive oropharyngeal cancer were nearly 90%, they dropped to about 76% in patients with stage T3 or T4 oropharyngeal cancer, demonstrating a continued need to improve therapeutics for later-stage oropharyngeal cancers [5]. In addition, long-term toxicities after combination chemoradiotherapy have been reported, including 25% and 47% of patients developing pharyngeal-laryngeal toxicity and oral cavity toxicity, respectively, at 10 years [7]. Another study reported gastrostomy tube dependence or treatment-related death in 43% of patients within 2–3 years, and discussions to reduce the intensity of standard treatments have appeared in the literature [8,9].

For early stage disease, the implementation of robotic-assisted surgical techniques [10,11] or elective neck dissection procedures has been shown to significantly reduce recurrence rates [12,13]. However, patients with node involvement have significantly lower progression-free survival and a higher risk of recurrence [14], and approximately 50% of patients with locally advanced HNSCC recur or develop metastatic disease after primary treatment [15]. Furthermore, tumor involvement in particular sites—such as the pterygoid muscles, skull base, and carotid arteries—precludes many standard therapies, limiting therapeutic options and resulting in poorer prognosis for patients [16]. Immune checkpoint inhibitors, including nivolumab and pembrolizumab, have prolonged median overall survival and reduced death rates compared with standard therapy for recurrent or metastatic HNSCC, but provide only a 1.5–2-month extension of median overall survival, resulting in relatively high death rates of 55–75% [17,18,19]. While the incidences of grades 3 and 4 treatment-related adverse events were lower and the quality of life reported by patients who received nivolumab was stable or slightly improved from baseline compared with patients given standard therapy, the modest improvements in median overall survival demonstrate the need for alternative therapeutic options for recurrent or locally advanced HNSCC.

Near-infrared photoimmunotherapy (NIR-PIT), which is a photo-activated cancer treatment based on the photo-induced ligand release reaction, is an emerging treatment modality which utilizes a light-activatable dye IRDye700Dx (IR700) conjugated to a monoclonal antibody (antibody–photoabsorber conjugate; APC) targeted against tumor-associated antigens located on the cell surface of tumor cells [20]. NIR light irradiation selectively destroys cells that have bound the APC. Within 1 min of NIR light irradiation, APC-bound cancer cells undergo cytotoxicity and subsequent cellular swelling, bleb formation, and vesicle rupture characteristic of immunogenic necrosis. Target specificity was confirmed with receptor-negative cells incubated with APC which did not undergo phototoxicity following NIR light irradiation. Bioluminescence imaging and ^18^F-FDG-PET confirmed necrosis of greater than 90% of cancer cells in vivo. Within 3 days of the NIR light irradiation, evidence of tumor size reduction was observed with no overt side effects in tumors which only received NIR light exposure. Tumor-free survival was observed in more than 80% of mice treated with combination therapy consisting of NIR-PIT with immune checkpoint inhibitors or NIR-PIT targeted against immunosuppressive cells. The fluorescence produced by the IR700 component of the APC delivers the light and also serves as a monitoring tool. Sufficient NIR light exposure can be confirmed with diminished IR700 fluorescence, thereby inducing maximum therapeutic effects. The results of a clinical trial of combination therapy of NIR-PIT with a checkpoint inhibitor showed a similar tendency in patient outcomes (NCT05265013).

Minimal cytotoxic effect of mAb-IR700 NIR-PIT was observed when APCs bound to the cell membrane and applied no NIR light when APCs are unbound because cytotoxicity is induced by the membrane damage through a photo-induced ligand release reaction that is different from conventional non-targeted or targeted photodynamic therapies that are based on reactive oxygen species. Over 20 different molecular targets expressed on various cancers have been utilized for several pre-clinical NIR-PIT studies. Targets include epidermal growth factor receptor (EGFR), human epidermal growth factor type 2, prostate-specific membrane antigen, CD25, Glypican-3, mesothelin, CD133, CD44, carcinoembryonic antigen, and more [21,22,23,24,25,26,27]. EGFR is a highly expressed antigen in many cancers, including HNSCC, and its levels of expression have been shown to correlate with disease severity; therefore, cancer therapeutics targeting EGFR, such as cetuximab, have been developed and proved efficacious [28,29,30]. By conjugating cetuximab to IR700, the experimental drug RM-1929 was developed with the potential to specifically target EGFR overexpressed in tumors such as HNSCC using the NIR-PIT paradigm. Preclinically, intravenous infusion of RM-1929 followed by NIR light has been shown to selectively damage or kill cell membranes of tumor cells and induce immunogenic cell death [31,32,33]. These observations led to Phase I and II testing, leading to a global Phase III trial. In September 2020, NIR-PIT using RM-1929 (RM-1929 NIR-PIT) was conditionally approved for the treatment of unresectable locally advanced or recurrent HNSCC by the Japanese Ministry of Health, Labor and Welfare. Since then, there have been a number of case series and clinical trial data published from Japan. Here, we present a review of the outcome data from the published case series of RM-1929 PIT for recurrent HNSCC.

## 2. Methods

### 2.1. Data Sources and Searches

A systematic search of PubMed and Embase was performed to identify clinical reports of RM-1929 NIR-PIT. The Preferred Reporting Items for Systematic Reviews and Meta-analyses (PRISMA) guidelines were utilized as a reference for this review. As this is a scoping review, this review was not registered with PRISMA. The following search terms were utilized: “RM-1929” OR “IR700” OR “Akalux” OR “AlluminoxTM” OR “Cet-IR700” OR “Cetuximab sarotalocan sodium” OR “Near-infrared photoimmunotherapy”. For PubMed, the following filters for article type were applied: “Case report”, “Clinical study”, “Clinical trial, Phase I”, “Clinical trial, Phase II”, “Clinical trial, Phase III”, “Clinical trial, Phase IV” and “Review”. For Embase, the following filters for study type and publication type were applied: “Clinical article”, “Phase 1 clinical trial”, “Clinical trial”, “Major clinical study”, “Phase 2 clinical trial”, “Phase 3 clinical trial”, “Case report”, “Clinical study”, “Case report”, “Article” and “Review”. Review articles were included to search for review articles which include newly reported patient cases. No language restrictions were applied. Authors were not contacted for missing data.

### 2.2. Study Selection

One investigator (NLM) performed the literature search and screened titles and abstracts. Studies included in the review had to meet the following criteria: (1) reported clinical cases of RM-1929 NIR-PIT with more than one patient, (2) reported prior therapies utilized and (3) reported parameters of the RM-1929 NIR-PIT. Review articles with no new patient cases and pre-clinical studies were excluded. Selected studies were cross-referenced to potentially identify additional studies not included in the initial search.

### 2.3. Data Extraction

One investigator (NLM) performed data extraction. Two additional investigators were consulted to resolve any discrepancies in the data collected. The following data were extracted from each article: patient demographics (gender and age), location of tumor recurrence, prior lines of therapy utilized, parameters of RM-1929 NIR-PIT (dose, laser parameters, diffuser type, number of cycles, etc.), treatment-emergent adverse events (TEAEs), complications and response rates when reported. Adverse events or complications were scored according to the National Cancer Institute-Common Terminology Criteria for Adverse Events (NCI-CTCAE) in 3 studies and were unspecified in the remaining studies. Response rates were based on the modified Response Evaluation Criteria in Solid Tumors (mRECIST 1.1) in 4 studies. Two studies included reports of patient quality of life measures and pain scores before and after RM-1929 NIR-PIT and were utilized in the discussion of qualitative patient data. One study did not report patient outcomes and was labeled “not specified” for the review. The study was included for the purpose of discussing pain management.

## 3. Results

### 3.1. Search Results

The primary search resulted in 37 articles from PubMed, 11 articles from Embase, and 2 articles from supplemental sources. After removing duplicates, 44 articles were screened by title and abstract. After screening, 39 articles were excluded for the following reasons: 30 articles were reviews with no new patient reports, 5 articles were case reports with only one patient, 3 articles reported pre-clinical results and 1 article reported a study that did not use RM-1929. After full-text review and cross-referencing, five studies were included in the final review (Figure 1). One of the studies included did not report patient outcomes but was incorporated into the review to shed light on pain management. All searches were performed on 7 June 2023.

### 3.2. Clinical Trial Data

Two publications reporting clinical trial data were identified. Cognetti et al. [34] reported outcomes of a Phase I/IIa clinical trial including 30 patients between 39 and 86 years old and predominantly male (80%) (Table 1). The majority of patients had tumor recurrence in the neck (43%), oral cavity (30%) and oropharynx (23%). All patients had previously undergone surgery and radiotherapy, while 70% received chemotherapy, 37% received immunotherapy and 27% received other therapies (Table 2). Patients received 1–4 treatments of RM-1929 NIR-PIT (640 mg/m^2^, physician’s choice of 50 J/cm^2^ for surface illumination with a frontal diffuser or 100 J/cm^2^ for interstitial illumination with a cylindrical diffuser), and outcomes were based on follow-up of 22.18–32.33 months and evaluated with the mRECIST 1.1 criteria.

Most patients achieved disease control (80%), with 4 (13%), 9 (30%) and 11 (37%) patients achieving complete response (CR), partial response (PR), and stable disease (SD), respectively (Table 1). The majority of TEAEs were lower grade, including edema (50%) and fatigue (33.3%) (Table 2). Higher-grade TEAEs were less common but still occurred in 19 patients and included anemia, dysphagia, localized pain, localized edema, hyponatremia and tumor hemorrhage. Three patient deaths were reported due to cervical vessel rupture, carotid artery hemorrhage and pneumonia 19–32 days after RM-1929 NIR-PIT. Given that the adverse events occurred over 19 days after treatment, the authors noted that the patient deaths were likely attributed to disease progression due to a lack of tumor response to therapy rather than the procedure itself.

Tahara et al. [35] reported outcomes of a smaller Phase I clinical trial including three patients who were all female. Tumor recurrence sites were reported in the neck, auditory canal and oropharynx. All patients previously underwent radiotherapy, chemotherapy, and alternative therapies (such as immunotherapy, hormonal therapy, or biologic therapy), and one patient received prior surgery. All patients received one treatment of RM-1929 NIR-PIT (640 mg/m^2^, physician’s choice of 50 J/cm^2^ for surface illumination with a frontal diffuser or 100 J/cm^2^ for interstitial illumination with a cylindrical diffuser) and outcomes were based on a 4-week follow up and evaluated with the mRECIST 1.1 criteria.

Two patients achieved disease control with a partial response. The remaining patient who had a recurrence in her auditory canal had progressive disease (PD) likely attributed to difficulties in effectively accessing portions of the tumor, which had infiltrated the bone. Similar to Cognetti et al., the majority of TEAEs reported were lower grade and commonly included localized pain and edema. One patient reported higher-grade transient localized pain.

### 3.3. Case Series and Retrospective Study Data

Nishikawa et al. [36] reported outcomes from a case series of 10 patients between 54 and 88 years old and predominantly male (90%). Twenty percent of patients had a recurrence in the cervical skin, oropharynx, or subcutaneous facial tissue, while the remaining patients had a recurrence in the oral cavity, glottis, and nasal cavity. Prior therapies utilized were only reported for two patients who both received surgery and either radiotherapy or chemoradiotherapy. Patients received 1–3 treatments of RM-1929 NIR-PIT (640 mg/m^2^, physician’s choice of 50 J/cm^2^ for surface illumination with a frontal diffuser or 100 J/cm^2^ for interstitial illumination with a cylindrical diffuser), and outcomes were evaluated with the mRECIST 1.1 criteria. Follow-up periods were reported for only two patients and ranged from 13 to 16 months. Patients achieved disease control with PR in 70% and CR in 30% of patients. Lower-grade TEAEs were pain (60%) and edema (40%). Higher-grade edema was reported in only one patient (10%).

Okamoto et al. [37] reported outcomes based on a retrospective study of nine patients between 67 and 77 years old, predominantly male (88.9%). Tumor recurrence in the oropharynx was most common (55.6%), while the remaining patients had recurrence in the oral cavity, maxillary sinus and cervical lymph node. All patients previously underwent surgery and radiotherapy and two patients had prior chemotherapy. Patients received 1–4 treatments of RM-1929 NIR-PIT (640 mg/m^2^, physician’s choice of 50 J/cm^2^ for surface illumination with a frontal diffuser or 100 J/cm for interstitial illumination with a cylindrical diffuser), and outcomes were based on 4-week follow-up and evaluated with the mRECIST 1.1 criteria.

At 4 weeks, all patients achieved disease control, with two achieving CR, six achieving PR and one patient with SD. Similar to the aforementioned studies, the majority of TEAEs reported were low grade pain (88.9%) and mucositis (77.8%). Higher-grade edema, dysphagia and hyponatremia were reported but with a lower frequency (22.2%). Patients continued to be followed for several months, during which four patients developed progressive disease after RM-1929 NIR-PIT, including distant metastases and extension to the carotid artery, and were switched to pembrolizumab. Pembrolizumab was not effective in one patient who subsequently died after supportive care.

### 3.4. Qualitative Patient Measures: Quality of Life and Pain Management

Two studies obtained additional measures on quality of life. Okamoto et al. described patient quality of life measures, while Shibutani et al. described patient pain levels and pain management [37,38]. In nine patients, quality of life across all functional and domain scales, assessed using the European Organization for Research and Treatment of Cancer (EORTC) Quality of Life Questionnaire Core 30 Module (QLQ-C30) and the Quality of Life Questionnaire Head and Neck Cancer Module (QLQ-H&N35), respectively, was not significantly different from baseline to 4 weeks after RM-1929 NIR-PIT (Table 3). While quality of life was not worsened by RM-1929 NIR-PIT, it was also not found to be significantly improved after treatment.

In five patients, reported pain levels, assessed using the Numerical Rating Scale (NRS), were highest on the day of NIR-PIT treatment and rapidly decreased across the following days [38] (Table 4). Comparisons between procedures performed with different diffuser types suggest higher pain levels in patients who underwent the procedure using the cylindrical diffuser as opposed to the frontal diffuser. Similar trends were observed with opioid use for pain management with higher maximum doses of fentanyl injections needed in patients for whom a cylindrical diffuser was utilized. In one patient, on whom the frontal diffuser was utilized, opioid management was determined to be non-indicated. In all other patients, the dose of opioids administered decreased to preoperative levels by postoperative day 4, suggesting that severe postoperative pain is short-lived and manageable with opioid therapy.

## 4. Discussion

Several clinical case series of RM-1929 NIR-PIT for recurrent and/or unresectable HNSCC were identified and compiled in this review, which, to our knowledge, is the first review that includes multiple studies. Preliminary data suggest positive clinical outcomes with RM-1929 NIR-PIT, with over two-thirds of each cohort achieving disease control. While there were several patients who achieved CR, the most common positive clinical response was PR. Post-treatment adverse events were predominantly lower grade with local pain and edema most frequently reported. Higher-grade localized pain and edema were less common. In one study, however, 19 patients were reported to have TEAEs higher than or equal to grade 3 [34]. While not reflected in the remaining studies, these outcomes suggest that higher-grade TEAEs are possible with RM-1929 NIR-PIT. Two studies also investigated changes in patient quality of life and pain levels throughout treatment and indicated no significant changes in quality of life or maximum pain levels on the day of RM-1929 NIR-PIT, which was effectively managed with opioid pain management with rapid improvement [37,38]. Higher reported pain levels in patients receiving RM-1929 NIR-PIT with the cylindrical diffuser compared with the frontal diffuser are most likely attributed to the differences in surgical technique. Cylindrical diffusers are utilized for deeper lesions and are delivered through a needle catheter which penetrates the tissue, while frontal diffusers are utilized for surface lesions and do not require tissue penetration. Given that patients reported peak pain levels on the day of the procedure, the authors suggest the importance of preemptive pain management strategies to minimize the patient’s pain burden, particularly if a more invasive cylindrical diffuser is used [38].

Despite advancements in therapeutics for unresectable and/or recurrent HNSCC, prognosis remains poor [39]. Given that all patients included in the studies had failed standard treatments, it is encouraging to observe overall response rates of 63.5% (43.3–100%) and disease control rates of 86.5% (66.7–100%) across studies. While the number of cases remains limited, these preliminary findings suggest there is a defined therapeutic benefit of RM-1929 NIR-PIT for patients with unresectable and/or recurrent HNSCC.

In addition to the case series reviewed here, several single-patient case reports also described the use of navigation systems or image-guided surgical techniques to perform RM-1929 NIR-PIT. Kishikawa et al. reported the use of intraoperative ultrasound to localize and guide insertion of the catheters for laser illumination for oropharyngeal cancer [40]. Ultrasound navigation was compatible with both cylindrical and frontal diffusers with no reported adverse complications. Koyama et al. reported the use of a SteathStaion Navigation System S7 and computed tomography (CT) guidance for recurrent maxillary sinus cancer [41]. Similarly, no adverse complications were reported. Okamoto et al. reported the use of the a Ear, Nose and Throat (ENT) navigation system and CT guidance for maxillary gingival carcinoma with no adverse postoperative complications reported [42]. Omura et al. reported the use of endoscopy-guided RM-1929 NIR-PIT treatment for nasopharyngeal cancer [43]. With at least a PR reported in most cases, RM-1929 NIR-PIT appears to be compatible and can be potentially enhanced with the implementation of existing and emerging navigation systems. In particular, preoperative imaging that is compatible with intraoperative navigation systems can be beneficial to precisely plan and monitor the positioning of the diffusers to ensure complete coverage of the lesion without significant overlap to decrease the likelihood of recurrence. Additionally, as reported in Cognetti et al., bone involvement of the tumor can also present challenges in penetrating the laser illumination effectively, and improvements to access difficult areas may arise from the adoption of various image-guided navigation systems. Therefore, in the future, we can anticipate advancements in more effective irradiation techniques and improved safety.

While the reported findings are promising, there are some limitations to this case-series review. In particular, follow-up periods to evaluate treatment response varied widely across studies, ranging from 4 weeks to 32.33 months. While treatment response timelines vary across patients, it is possible that shorter follow-up periods may fail to capture recurrences which can be found with longer follow-up periods. In addition, due to the relatively short follow-up periods, there is little understanding of the potential long-term adverse effects which have been described in other standard treatments. For instance, cisplatin–radiotherapy combination therapy for HPV-positive oropharyngeal cancer has been associated with long-term pharyngeal–laryngeal or oral cavity toxicity at 10 years post treatment [7].

Several single-patient case reports were not included, as response rates could be easily skewed, which limited the number of studies included in this review. In addition, only two studies included a discussion of pain management and pain scores. While the preliminary reports suggest a relatively high safety profile, additional studies with longer follow-up periods are needed to draw clear conclusions regarding the safety profile and pain management approach compatible with RM-1929 NIR-PIT. In particular, delineating how certain parameters of the procedure tend to correlate with increased patient pain can potentially help inform surgical planning and preemptive pain management strategies. Current preliminary data suggest that increasing the number of cylindrical diffusers may lead to increased postoperative pain. There is, however, no data on how the number of diffusers utilized in a single procedure or the number of target lesions impact postoperative pain. Navigation systems may not only improve irradiation techniques but also shed light on more effective and potentially less invasive approaches to further improve postoperative pain. As additional clinical studies continue to arise, the inclusion of pain measures in addition to clinical outcomes and adverse complications can provide a more holistic understanding of the efficacy of RM-1929 NIR-PIT.

In addition to technical procedures for avoiding overexposure to light, based on the photochemical mechanism, we propose the pre-treatment of patients with ascorbic acid, a strong reducing agent, and a proton donor. Since the acute inflammatory reaction accompanying edema and pain, which are major adverse side effects after NIR-PIT, can be induced by reactive oxygen species produced mainly by the exposure of laser light to unbound APC, such reactive oxygen species can be quenched by reducing agents, including ascorbic acid. Ascorbic acid also functions as an electron and proton donor, promoting the photo-induced ligand release reaction of IR700, which is the major cause of selective cytotoxicity induced by NIR-PIT [44]. Therefore, theoretically, the pre-administration of ascorbic acid prior to laser light exposure could suppress acute inflammatory side effects, including acute edema and pain, without compromising or even further promoting the selective cytotoxicity of NIR-PIT. Indeed, ascorbic acid pre-administration before light exposure successfully suppressed acute edema but did not affect the therapeutic outcome of NIR-PIT, as shown in both immunocompromised and immunocompetent mouse tumor models in the preclinical study [44]. Although pain was difficult to measure in animal experiments, the mice appeared comfortable without acute edema after NIR-PIT. Since ascorbic acid has demonstrated good safety profiles even with massive administration for cancer treatment, clinical testing is pending, as edema and pain are the most frequently observed adverse side effects of NIR-PIT, as described above (NCT02905578, NCT02905591, NCT04046094).

The effects of ascorbic acid on cytotoxicity induced by NIR-PIT and acute edema formation are clear evidence that distinguishes NIR-PIT from conventional or even targeted photodynamic therapy (conventional photoimmunotherapy, PDT), where therapeutic effects rely on reactive oxygen species production and can be quenched with ascorbic acid. NIR-PIT is designed to cause cytotoxicity through cellular membrane damage selectively on target APC-bound cells based on the photo-induced ligand release reaction. Therefore, to maintain pharmacokinetics and specific binding, APC is designed to be hydrophilic, keeping APC outside of cells. Since ascorbic acid is also a hydrophilic and charged molecule, all reducing and protonation reactions occur in the extracellular space, resulting in the expected effects as hypothesized. The agents used for antibody-targeted PDT, referred to as photoimmunotherapy, are designed with the same principles as NIR-PIT, but their cytotoxicity relies on reactive oxygen species. Consequently, reactive oxygen species are produced near, but outside, the cellular membrane, where they can be quenched by ascorbic acid. In contrast, conventional PDT agents, which were used in targeted PDT for conjugating with an antibody in the past, are hydrophobic and designed to permeate into cells. When exposed to light during PDT, reactive oxygen species are produced inside the cells and mainly affect the electron transfer system to activate the caspase pathway, inducing cytotoxicity, although PDT cytotoxicity is non-specifically induced with other different causes, including heat and cell membrane damage, etc. Therefore, ascorbic acid only partially affects PDT cytotoxicity. These differing reactions to ascorbic acid clearly illustrate the differences in the chemical basis between NIR-PIT and PDT.

## 5. Conclusions

RM-1929 NIR-PIT appears to be a promising therapeutic option for patients with unresectable and/or recurrent HNSCC reflected in the favorable disease control rate observed across studies. Preliminary findings suggest NIR-PIT has a minimal impact on quality of life, and pain levels are manageable using existing opioid pain management approaches. As the multi-institution Phase III clinical trial (NCT03769506) continues and additional case reports/series in Japan are published, further systematic reviews of RM-1929 NIR-PIT clinical efficacy are warranted in the future.

## Figures and Tables

**Figure 1 cancers-15-05117-f001:**
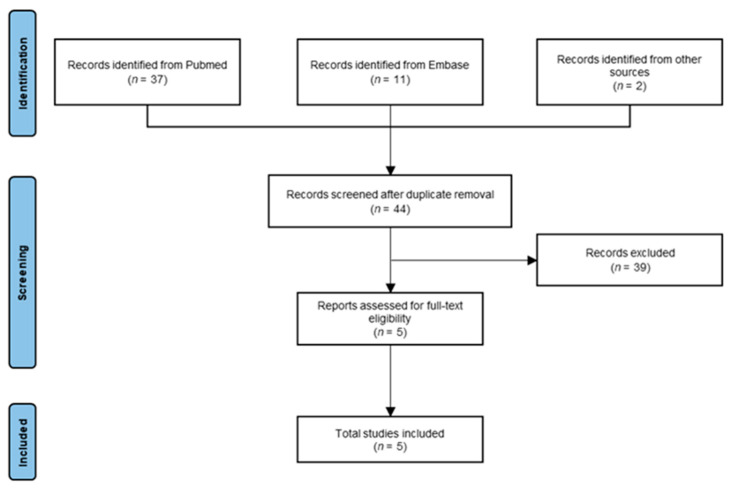
PRIMSA flowchart of study selection.

**Table 1 cancers-15-05117-t001:** Patient outcomes and demographics.

Author	Year	Design	# of Patients	Age Range	Male (*N*)	Female (*N*)	Diagnosis	Tumor Recurrence Sites (*N*)	CR (%)	PR (%)	SD (%)	DC (%)	PD (%)	ORR (%)
**Cognetti et al.** [34]	2021	Phase I/IIa clinical trial	30	39–86	24	6	HNSCC	Neck (13) Oral cavity (9) Oropharynx (7) Skin (3) Hypopharynx (2) Sinus (2) Nasal cavity (1) Nasopharynx (1) Parotid gland (1) Occipital gland (1)	4 (13.3)	9 (30.0)	11 (36.7)	24 (80)	6 (20.0)	13 (43.3)
**Tahara et al.** [35]	2021	Phase I clinical trial	3		0	3	HNSCC	Soft tissue in mental/submental region (1) External auditory canal (1) Superficial cervical node (1) Oropharynx (1)	0 (0.0)	2 (66.7)	0 (0.0)	2 (66.7)	1 (33.3)	2 (66.7)
**Nishikawa et al.** [36]	2022	Review with case report	10	54–88	9	1	HNSCC	Cervical skin (2) Oropharynx (2) s.c. facial tissue (2) Glottis (1) Tongue (1) Lower gingiva (1) Nasal cavity (1)	3 (30.0)	7 (70.0)	0 (0.0)	10 (100)	0 (0.0)	10 (100)
**Okamoto et al.** [37]	2022	Retrospective study	9	67–77	8	1	HNSCC	Oropharynx (5) Buccal mucosa (1) Tongue/upper and lower gingiva (1) Maxillary sinus (1) Cervical lymph node (1)	2 (22.2)	6 (66.7)	1 (11.1)	9 (100)	0 (0.0)	8 (88.9)
**Shibutani et al.** [38]	2023	Case report	5	51–74	2	3	HNSCC	Buccal mucosa (2) Oropharynx (2) Nasopharynx (1)	n/a	n/a	n/a	n/a	n/a	n/a
**Overall**			57	39–86	43	14	HNSCC		9 (17.3)	24 (46.2)	12 (23.1)	45 (86.5)	7 (13.5)	33 (63.5)

Abbreviations: HNSCC, head and neck squamous cell carcinoma; CR, complete response; PR, partial response; SD, stable disease; DC, disease control; PD, progressive disease; ORR, overall response rate.

**Table 2 cancers-15-05117-t002:** Treatment-emergent adverse events.

Author	Year	Prior Therapies (*N*)	NIR-PIT Parameters	Tumor Response Evaluation Method	Follow-Up Period	TEAE Evaluation	Adverse Events < Grade 3 (*N*)	Adverse Events > Grade 3 (*N*)	Patient Deaths (*N*)
**Cognetti et al.** [34]	2021	Surgery (30) Radiotherapy (30) Chemotherapy (21) Immunotherapy (11) Biological/hormonal/other (8)	640 mg/m^2^ 1–4 cycles	mRECIST 1.1	22.18–32.33 mo	NCI CTCAE 4.03	Edema (15) Fatigue (10) Dysphagia (7) Constipation (6) Erythema (6) Anemia (5) Dehydration (5) Facial pain (4) Oral pain (4) Tumor pain (4) Local swelling (4) Cough (4) Oropharyngeal pain (4) Pneumonia (4) Weight loss (4)	Anemia (3) Dysphagia (2) Local pain (2) Oral pain (2) Local edema (2) Hyponatremia (2) Hemorrhage (2) Tumor pain (2) Pneumonia (2) Edema (1)	3
**Tahara et al.** [35]	2021	Surgery (1) Radiotherapy (3) Chemotherapy (3) Immuno/hormonal/biologic/other therapy (3)	640 mg/m^2^ 1 cycle	mRECIST 1.1	4 wks	NCI CTCAE 4.30	Local pain (2) Local edema (1) Facial edema (1) Elevated BP (1) Elevated GGT (1) Decreased WBC (1) Anemia (1) Glossitis (1) Abnormal hepatic function (1) Generalized rash (1)	Local pain (1)	0
**Nishikawa et al.** [36]	2022	Surgery (2) Radiotherapy (1) Chemoradiotherapy (1) Not reported (8)	640 mg/m^2^ 1–3 cycles	mRECIST 1.1	13–16 mo	Not specified	Pain (6) Hemorrhage (1) Edema (4) Fistula (3)	Edema (1)	0
**Okamoto et al.** [37]	2022	Surgery (9) Radiotherapy (9) Chemotherapy (2)	640 mg/m^2^ 1–4 cycles	mRECIST 1.1	4 wks	NCI CTCAE 4.30	Pain (8) Mucositis (7) Edema (4) Nausea (3) Hemorrhage (2) Diarrhea (2) Dysphagia (1) Rash (1) Fever (1) Aspiration (1) Hyperkalemia (1) Trismus (1) Constipation (1) Dehydration (1) Intratumoral broken needle fragment (1) Oral dysesthesia (1)	Edema (2) Dysphagia (2) Hyponatremia (2) Pain (1) Mucositis (1) Acute kidney injury (1) Anemia (1) Hypokalemia (1) Liver dysfunction (1) Weight loss (1)	1
**Shibutani et al.** [38]	2023	Not specified	1–3 cycles	Not specified	4 d	Not specified	Not specified	Not specified	0

Abbreviations: mRECIST, modified Response Evaluation Criteria in Solid Tumors; NCI CTCAE, National Cancer Institute Common Terminology Criteria for Adverse Events.

**Table 3 cancers-15-05117-t003:** Quality of life scores from Okamoto et al. [37].

Measure	QOL Score Change from BASELINE Least Square Mean (95% CI)	*p*-Value v. Baseline
** *Functional scales* **		
**Physical functioning**	2.2 (−2.9–7.3)	0.347
**Role functioning**	5.6 (−12.6–23.7)	0.500
**Emotional functioning**	−3.7 (−10.9–3.5)	0.272
**Cognitive functioning**	0.0 (−11.1–11.1)	>0.999
**Social functioning**	3.7 (−10.3–17.7)	0.559
** *Global health status* **	−7.4 (−20.8–6)	0.237
** *Domain scales* **		
**Pain**	4.6 (−6.5–15.8)	0.366
**Swallowing**	−5.6 (−31.0–19.9)	0.628
**Sense problems**	−1.9 (−6.1–2.4)	0.347
**Speech problems**	0.0 (−26.7–26.7)	>0.999
**Trouble with social eating**	−6.5 (−17.5–4.5)	0.211
**Trouble with social contact**	6.7 (−6.1–19.5)	0.264
**Less sexuality**	−5.6 (−14.6–3.5)	0.195

Abbreviations: QOL, quality of life; 95% CI, 95% confidence interval.

**Table 4 cancers-15-05117-t004:** Pain scores and mean opioid doses from Shibutani et al.

	Mean NRS Pain Score (Mean Opioid Dose (mg))						
Diffuser Type	POD-2	POD-1	POD0	POD1	POD2	POD3	POD4
**Cylindrical**	5.3 (45)	5 (45)	7.8 (94)	4.6 (60)	4.2 (66)	3.8 (60)	4.8 (48)
**Frontal**	0 (0)	0 (0)	6.8 (46.3)	3 (56.3)	2.8 (0)	1.8 (0)	1.5 (0)

Abbreviations: NRS, numerical rating scale; POD, postoperative day.

## Data Availability

The data presented in this study are available in this paper.

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
