# Peer review of "Review of RM-1929 Near-Infrared Photoimmunotherapy Clinical Efficacy for Unresectable and/or Recurrent Head and Neck Squamous Cell Carcinoma"

_cancers, 2023, doi:10.3390/cancers15215117_

Round 1

Reviewer 1 Report

In this manuscript, the authors collected a series of case reports and clinical trial data to assess the efficacy of RM-1929 photoimmunotherapy (PIT). This preliminary data in real world use of RM-1929 PIT shows that it is a well-tolerated therapy that has clinically meaningful outcomes for patients with unresectable and/or recurrent HNSCC. However, some major concerns should be addressed.

(1) The content of the simple summary was identical to that of the abstract, which making simple summary overly detailed in this manuscript. The author had better cohesively summarize the current status of RM-1929 photoimmunotherapy clinical efficacy for unresectable and/or recurrent head and neck squamous cell carcinoma and the work of this manuscript in simple summary.

(2) The manuscript provided only a listing of cases in different papers and there was no analysis of them, which makes the manuscript insufficiently insightful and informative.

(3) The flow chart of studies for meta-analysis in Figure 1 should be more detailed and specific. For example, it is advisable to specify the number of patients in the studies (see for example European Heart Journal 2012, 33,1750-1757).

(4) The authors simply listed the data from the articles that have been reported and did not summarize and organize them in Tables 1 and Table 2. The data in Tables 1 and Table 2 were abundant but messy.

Author Response

(1) The content of the simple summary was identical to that of the abstract, which making simple summary overly detailed in this manuscript. The author had better cohesively summarize the current status of RM-1929 photoimmunotherapy clinical efficacy for unresectable and/or recurrent head and neck squamous cell carcinoma and the work of this manuscript in simple summary.

(Response) We have edited the short summary as the reviewer suggested.

(2) The manuscript provided only a listing of cases in different papers and there was no analysis of them, which makes the manuscript insufficiently insightful and informative.

(Response) We have added the overall number in Table 1 and add those numbers in the discussion section.

(3) The flow chart of studies for meta-analysis in Figure 1 should be more detailed and specific. For example, it is advisable to specify the number of patients in the studies (see for example European Heart Journal 2012, 33,1750-1757).

(Response) Due to the small case number of the studies, we were not categorized this article as a meta-analysis but a brief review article. The chart was added by the request of the managing editor. The number of patients in each study is included in Table 1.

(4) The authors simply listed the data from the articles that have been reported and did not summarize and organize them in Tables 1 and Table 2. The data in Tables 1 and Table 2 were abundant but messy.

(Response) We have added the overall numbers Table 1 and added the number in the discussion section.

Reviewer 2 Report

cancer-2575810-peer-review-v1

 This article is a good-written review article with concise presentation, logical analysis, and discussion. It can be accepted as current form. However, I would only provide one minor suggestion as followed:

1. In line 63, EGFR is a highly expressed antigen in many 63 cancers including HNSCC and levels of expression have been shown to correlate with disease severity, therefore cancer therapeutics targeting EGFR, such as cetuximab, have been developed and proved efficacious [15–17]”, the full name of EGFR should exist at the first time for most readers.

Author Response

  1. In line 63, EGFR is a highly expressed antigen in many 63 cancers including HNSCC and levels of expression have been shown to correlate with disease severity, therefore cancer therapeutics targeting EGFR, such as cetuximab, have been developed and proved efficacious [15–17]”, the full name of EGFR should exist at the first time for most readers.

 (Response) We have added the full name of EGFR as the reviewer suggested,

Reviewer 3 Report

The manuscript entitled Review of RM-1929 photoimmunotherapy clinical efficacy for unresectable and/or recurrent head and neck squamous cell carcinoma is a well written review on the clinical use of cetuximab-IRDye700dx.

First of all, from a perspective of the photodynamic therapy field, this is a milestone concerning the first testing in the clinic of antibody-targeted photosensitizers. Authors are kindly requested to give a small overview of the most recent efforts and challenges in respect of PDT usage in head and neck cancers. That information seems critical to put the results here described in perspective / in the appropriate context.  

Photodynamic therapy is a keyword that seems relevant to add

Please explain APC or reconsider its use.

This therapeutic approach selectively destroys cells which have bound the APC after exposure to near infrared (NIR) light irradiation. Please provide an accurate description of photodynamic therapy, including the production of reactive oxygen species and possible mechanisms of cell death.

PIT paradigm : it seems appropriate to refer to the first studies on PIT (1983, 1989,.. and a few examples of reviews)

https://pubmed.ncbi.nlm.nih.gov/6185591/

https://pubmed.ncbi.nlm.nih.gov/4028022/

https://pubmed.ncbi.nlm.nih.gov/2717640/

https://pubmed.ncbi.nlm.nih.gov/31783651/

https://pubmed.ncbi.nlm.nih.gov/14706444/

Patients received 1–4 cycles of RM-1929 PIT (640 mg/m2, 50 J/cm2 or 100 J/cm): what does 1-4 cycles mean? In the same intervention 4 different illuminations? Or 4 separate treatments? How much time apart? 50J/cm2 or 100J/cm? should be cm2? How was the decision made? Which W/cm2 were used? Or how long was each illumination? Was it with 1 optical fiber inserted in the tissue (interstitial) or was it superficial illumination? Wavelength should be added, laser used could be added.

Cylindrical or frontal diffuser: which treatments used which diffusers?

In all other patients, the dose of opioids administered decreased to preoperative levels postoperative day 4: perhaps at by or at postoperative day 4 for clarity  

Author Response

First of all, from a perspective of the photodynamic therapy field, this is a milestone concerning the first testing in the clinic of antibody-targeted photosensitizers. Authors are kindly requested to give a small overview of the most recent efforts and challenges in respect of PDT usage in head and neck cancers. That information seems critical to put the results here described in perspective / in the appropriate context.  

 (Response) We have added the recent new therapeutic choices including ICIs in the Introduction as suggested.

Photodynamic therapy is a keyword that seems relevant to add

 (Response) In order not to confuse the readers, we changed the term from “PIT” to “NIR-PIT” throughout the article. Additionally, we have added a paragraph to explain the difference between conventional PDT and NIR-PIT in the Discussion.

Please explain APC or reconsider its use.

 (Response) We have defined APC.

This therapeutic approach selectively destroys cells which have bound the APC after exposure to near infrared (NIR) light irradiation. Please provide an accurate description of photodynamic therapy, including the production of reactive oxygen species and possible mechanisms of cell death.

PIT paradigm : it seems appropriate to refer to the first studies on PIT (1983, 1989,.. and a few examples of reviews)

https://pubmed.ncbi.nlm.nih.gov/6185591/

https://pubmed.ncbi.nlm.nih.gov/4028022/

https://pubmed.ncbi.nlm.nih.gov/2717640/

https://pubmed.ncbi.nlm.nih.gov/31783651/

https://pubmed.ncbi.nlm.nih.gov/14706444/

  (Response) In order not to confuse the readers, we changed the term from “PIT” to “NIR-PIT” throughout the article. Additionally, we have added a paragraph to explain the difference between conventional PDT or antibody-targeted PDT that has never been used in the oncology clinic and NIR-PIT.

Patients received 1–4 cycles of RM-1929 PIT (640 mg/m2, 50 J/cm2 or 100 J/cm): what does 1-4 cycles mean? In the same intervention 4 different illuminations? Or 4 separate treatments? How much time apart? 50J/cm2 or 100J/cm? should be cm2? How was the decision made? Which W/cm2 were used? Or how long was each illumination? Was it with 1 optical fiber inserted in the tissue (interstitial) or was it superficial illumination? Wavelength should be added, laser used could be added.

Cylindrical or frontal diffuser: which treatments used which diffusers?

In all other patients, the dose of opioids administered decreased to preoperative levels postoperative day 4: perhaps at by or at postoperative day 4 for clarity  

  (Response) We have added the suggested details in the text.

Reviewer 4 Report

Comments to authors:

The manuscript by Miyazaki et al. provides a summary of 5 clinical studies involving NIR-PIT. The manuscript is interesting and summarizes key findings from these clinical studies. There are few suggestions, as provided below, which can help in enhancing the quality of the manuscript.

 1.       Discussion in lines 344-378 suggests that PIT also works through ROS generation, although it may be extracellular as compared to conventional PDT where ROS generation is expected to be mainly intracellular. However, in the introduction the authors suggest that NIR-PIT does not necessarily involve ROS. The authors should clarify these statements. Although it is agreed that ROS generation may be a by-product and not the main effector molecules leading to phototoxicity, these statements should be clearl articulated for a better understanding.

2.       Introduction lines, 76-78 and 96-99 are repetitive and the description of NIR-PIT should be removed from one of the two places.

3.       In the discussion, around lines 350-360, where the authors discus it might be appropriate to cite clinical trials involving NIR-PIT with pembrolizumab (NCT05265013).

4.       The statement that conventional PDT utilizes hydrophobic photosenstizers may not be correct. There are several hydrophilic derivatives of photosensitizers now available that rely on cellular internalization for their phototoxic effect.

5.       Disrupting the ETC (lines 375-377) is just one of the several ways in the conventional PDT may exert its cytotoxic effects. ER stress, damaging membrane proteins, etc can also be involved.

Author Response

  1. We have clarified the chemistry basis in 2 separate ACS journal articles ref#32 and #44.
  2. We have deleted the repetition.
  3. We have added the study number suggested by the reviewer at lines 95-96.
  4. We have clarified this better.
  5. We have added suggested causes of cytotoxicity induced by PDT.

Reviewer 5 Report

Miyazaki et al have reviewed the available clinical data for the use of antibody-IRDye700 conjugates (RM-1929) as near-infrared photoimmunotherapy agents. While the number of studies included in the review is small (5), the authors have produced a useful document that will help the reader grasp the clinical benefits and be aware of the safety and potential pitfalls of such novel and potentially important cancer treatments. The main weakness of the manuscript is the limited data available and the low patients' follow-up time, that preclude  definitive assessments on the therapy's efficacy and long-term effects. This is well acknowledged by the authors.
My main concern is that authors stress several times that the mechanism of action of RM-1929 is different from conventional photoimmunotherapy, in that ligand photorelease, as opposed to reactive oxygen photogeneration, is the responsible for cell death. Such a mechanistic discussion does not belong to this review and would therefore recommend to remove it. Other than that, this is a timely and welcome review. Time will tell how useful it will be for the advancement of photoimmunotherapy.

Author Response

(Response) We have eliminated the repetition as much as we could. The suggested description of the functional mechanism of NIR-PIT has been raised and suggested by Reviewer #3 and 4, therefore, we have added the current contents.

Round 2

Reviewer 1 Report

I suggest accepting the revised manuscript for publication.

Author Response

We are happy to read your positive comments. Thank you for reviewing this article. 

Reviewer 3 Report

The authors have addressed most of the points raised. There is still a point of concern:

Near-infrared photoimmunotherapy (NIR-PIT), which is a photo-activated cancer treatment based on the photo-induced ligand release reaction, yet does not rely on oxygen/reactive oxygen species that is different from conventional photodynamic therapies, : this is incorrect. The IRDye700DX has been shown to induce ROS and singlet oxygen production, thus the mechanism suggested above does not make sense.

see references: 

https://www.ncbi.nlm.nih.gov/pmc/articles/PMC3233641/ -> from the same authors, NaN3 could reduce the effect

https://www.ncbi.nlm.nih.gov/pmc/articles/PMC7116242/ -> the singlet oxygen quantum yield was determined for IRDye700DX alone

https://www.degruyter.com/document/doi/10.1515/nanoph-2021-0195/html?lang=en -> also here NaN3 was able to reduce the effect observed thus confirming ROS are involved.

Maximum efficacy of mAb-IR700 NIR-PIT was observed when APCs bound to the cell membrane and induced no phototoxicity when unbound because cytotoxicity is induced by photo-induced ligand release reaction that is different from conventional non-targeted or targeted photodynamic therapies that based on reactive oxygen species. : again the PS is a phthalocyanine derivative from a well known class of photosensitizers that has a singlet oxygen quantum yield. The explanation for not working when unbound is because the ROS produced upon illumination are very short lived and do not really diffuse, thus there is only cytotoxicity produced when the antibody is bound and thus the PS is close to the membrane or already inside cells. Authors should correct these 2 sections and acknowledge the PDT field. 

Author Response

Please read our more recent chemistry articles published in American Chemical Society journals below cited as #32 and #44 to fully understand this new chemistry of IR700 that is different from conventional PDT relying on ROS. In response to the first and third questions, we also described the chemical function of NaN3 that is not working as a dedicated singlet oxygen quencher written in the old chemistry text books but working on IR700 as N3 radical to quench IR700 anion radical that led to inhibit photo-induced ligand release of IR700 in the second article. We described this new chemistry with full evidences including electron transfer with ESR as written in the second paper. Therefore, ACS professional folks accepted this photo-chemical mechanism of IR700 that we described in this paper.

  1. Sato K, Ando K, Okuyama S, Moriguchi S, Ogura T, Totoki S, Hanaoka H, Nagaya T, Kokawa R, Takakura H, Nishimura M, Hasegawa Y, Choyke PL, Ogawa M, Kobayashi H. Photo-induced ligand release from a silicon phthalocyanine dye conjugated with monoclonal antibodies; A mechanism of cancer cell cytotoxicity after near infrared photoimmunotherapy. ACS Central Science 2018 ;4(11):1559-1569.
  1. Kato T, Okada R, Goto Y, Furusawa A, Inagaki F, Wakiyama H, Furumoto H, Daar D, Turkbey B, Choyke PL, Takakura H, Inanami O, Ogawa M, Kobayashi H. Electron Donors rather than Reactive Oxygen Species Need for Therapeutic Photochemical Reaction of Near Infrared Photoimmunotherapy. ACS Pharmacology and Translational Science 2021; 4(5): 1689–1701.

Round 3

Reviewer 3 Report

The 2 papers indicated show a mechanism by which the PS is modified upon illumination, losing the sulfonate chains that provide more water solubility. There is however no direct proof that the aggregated PS is in fact leading to the cytotoxicity, that is only the mechanism proposed, since the reduction of fluorescence of the PS can also be explained by the photobleaching of the PS (much more common process, of ring opening).  

In view of the recent work by the authors (again only proving the PS can be found without the side chain/ or the ligand as called, but possibly just a part of it), but certainly taking in account other work, including from the same authors (e.g. https://www.ncbi.nlm.nih.gov/pmc/articles/PMC4508222/) and others who have detected ROS production from IRDye700DX illumination, and many other PS experts who have worked with Pc before, the request now is to instead of excluding the ROS production to simply acknowledge both mechanisms. There is simply no direct proof that can really exclude the ROS production in vitro and in vivo, while several studies have shown it can be measured. Thus, both could co-exist, which would seem much more realistic than that the proposed mechanism is the solely occurring mechanism. Also, studies have shown that NaN3 used to quench single oxygen have generally reduced but not eliminated the effects. This could indeed imply other mechanisms besides singlet oxygen are involved. 

Thus sentences such as: "Near-infrared photoimmunotherapy (NIR-PIT), which is a photo-activated cancer treatment based on the photo-induced ligand release reaction, yet does not rely on oxygen/reactive oxygen species that is different from conventional photodynamic therapies" should be corrected.  Also the discussion should be adapted.  

Author Response

(Response) I’m sorry to say that the reviewer misread the chemistry in the cited papers. “the PS is modified upon illumination, losing the sulfonate chains that provide NO water solubility.” That is the reason why APC made aggregation.

The suggested Kishimoto’s old article is truly controversial one. The authors did not follow the original procedure and made two totally different results, but they like to publish one of these as conclusion. Therefore, our group initially taught the first author, but all our team in Laboratory of Molecular Theranosticsin the MIB/NCI that invested this technology declined the authorship.We still did not know why the authors chose the one result and discarded other. Therefore, we published to clarify the chemical background in 2018 and 2021 papers.

Taken together, the comment that the reviewer suggested to edit is correct.